# Cartographic Vandalism in the Era of Location-Based Games—The Case of OpenStreetMap and Pokémon GO

**Levente Juhász** [1,*] **, Tessio Novack** [2] **, Hartwig H. Hochmair** [3] **and Sen Qiao** [4]

1   GIS Center, Florida International University, Miami, FL 33199, USA
2   GIScience Research Group, Heidelberg University, 69117 Heidelberg, Germany; novack@uni-heidelberg.de
3   Geomatics Program, University of Florida, Fort Lauderdale, FL 33314, USA; hhhochmair@ufl.edu
4   Department of Computer Science, North Carolina State University, Raleigh, NC 27695, USA; sqiao@ncsu.edu
*   Correspondence: ljuhasz@fiu.edu; Tel.: +1-305-348-6444

**Abstract:** User-generated map data is increasingly used by the technology industry for background mapping, navigation and beyond. An example is the integration of OpenStreetMap (OSM) data in widely-used smartphone and web applications, such as Pokémon GO (PGO), a popular augmented reality smartphone game. As a result of OSM's increased popularity, the worldwide audience that uses OSM through external applications is directly exposed to malicious edits which represent cartographic vandalism. Multiple reports of obscene and anti-semitic vandalism in OSM have surfaced in popular media over the years. These negative news related to cartographic vandalism undermine the credibility of collaboratively generated maps. Similarly, commercial map providers (e.g., Google Maps and Waze) are also prone to carto-vandalism through their crowdsourcing mechanism that they may use to keep their map products up-to-date. Using PGO as an example, this research analyzes harmful edits in OSM that originate from PGO players. More specifically, this paper analyzes the spatial, temporal and semantic characteristics of PGO carto-vandalism and discusses how the mapping community handles it. Our findings indicate that most harmful edits are quickly discovered and that the community becomes faster at detecting and fixing these harmful edits over time. Gaming related carto-vandalism in OSM was found to be a short-term, sporadic activity by individuals, whereas the task of fixing vandalism is persistently pursued by a dedicated user group within the OSM community. The characteristics of carto-vandalism identified in this research can be used to improve vandalism detection systems in the future.

**Keywords:** volunteered geographic information; OpenStreetMap; vandalism; Pokémon; location-based games; user behavior analysis

## 1. Introduction

*1.1. Motivation*

Advancements in information and geospatial technologies have triggered significant changes in spatial data creation and consuming behavior throughout the last few decades. The participation of the public in the creation of geodata has been captured by a number of different terms in the literature, such as volunteered geographic information (VGI), crowdsourcing, user generaged geographic content (UGGC) and citizen science. As opposed to traditional spatial data creation, at least part of this process relies on citizens who have various levels of expertise. Community-based data collection platforms oftentimes also lack quality protocols and standards, so that the data quality of participatory spatial data may become a concern to its users [1].

Despite this potential drawback, participatory map-making has gained in popularity. In recent years, several VGI platforms emerged that were built on top of the idea of volunteerism and open data sharing. The collaborative nature of VGI mapping projects allows anyone to contribute to the creation and improvement of spatial databases through adding, modifying or deleting map features. These projects often publish their data under open licenses that allow third parties to use and build upon these maps. OpenStreetMap (OSM) is a prominent VGI project [2] published under the Open Database License (ODbL) (https://opendatacommons.org/licenses/odbl/index.html), which allows the utilization of its data by third parties. Mainstream technology companies have begun to use OSM data in recent years. Two prominent examples are Snapchat and Pokémon GO (PGO), which both became OSM data consumers recently. Snapchat reported 218 million daily active users in 2019 [3]. PGO was used by 28.5 million players every day during its peak popularity in 2016 and it still managed to engage more than 10 million monthly users in 2018 [4]. Similarly, widely popular mapping and navigational services, such as Google Maps and Waze also implemented crowdsourcing components to keep their map data up-to-date. The large user base of these applications puts a spotlight on geospatial data, and on VGI in particular, which can be seen as a validation of the concept, as it proves that citizens are able and willing to build massive map databases and feed geospatial services with accurate information. However, increased attention comes with undesirable side effects. That is, participatory geospatial projects are vulnerable to cartographic vandalism, which is defined as a defiant behavior directed at geospatial data [5]. Since UGGC today is viewed by hundreds of millions of users through online applications, vandalism [6] no longer stays within mapping communities but is visible to a worldwide audience. This increased visibility of vandalized content (e.g., fake place names, fictional data) threatens to undermine the reputation of collaborative mapping projects. Several cases of cartographic vandalism took place in recent years. For example, in August 2018, a case of anti-semitic vandalism surfaced on Snapchat's online maps [7] that was based on OSM. This incident made it to various popular media outlets, such as the BBC, Time and The New York Times, and therefore showed collaboratively generated spatial data in a bad light. Vandalism can also be observed in connection with location-based games, though that type of vandalism is usually not motivated by hatred or prejudices against a specific group of people. Instead, in PGO, users were found to modify the underlying OSM data by adding fictional map features (e.g., parks, footpaths and lakes) to gain benefits in the game [8]. Cartographic vandalism can be problematic in many ways. First, the undermined credibility of data generated by participatory projects may limit the future growth and reach of the crowdsourcing platform. Second, if shared data are already used in operational systems, such as navigational services, vandalized content may directly affect users by putting them in dangerous traffic situations or misguiding them. Another problematic aspect is related to a widely accepted advantage of collaborative map data; namely, its timeliness. While the interaction between authorities and participatory approaches of collecting spatial data is complex due to the different objectives of these entities [9], it was found that in some instances, such as following natural and man-made disasters or other emergency situations, the public can be a reliable data source. Moreover, it is often the only data source that provides updated data within a short time. A well-known showcase of this type of public engagement is the maps created by the Humanitarian OpenStreetMap Team (HOT) that were used by first responders to save lives after the Haiti earthquake in 2010 [10]. It is therefore an open question if authorities will continue to use geographic data based on participatory approaches, such as VGI, when trust in the data is undermined by obvious cases of cartographic vandalism. From the perspective of the volunteer mapping community, discovering and fixing malicious content takes significant time. This time could, however be better spent on improving the maps in other important tasks, such as mapping missing areas.

As outlined above, the vulnerability to vandalism is often considered one of the drawbacks of participatory spatial data projects. However, it has been addressed only sporadically in the literature so far. This research therefore aims to contribute to a better understanding of cartographic vandalism in the context of location-based games. Using PGO and OSM as analysis platforms, it aims to describe

the effect and nature of cartographic vandalism at the data level. It also analyzes the mapping community's response to vandalism. More specifically, the objectives of this study are to:

- Develop a method for the collection of a large sample of PGO-related cartographic vandalism events and their fixes in OSM.
- Analyze the temporal dynamics of cartographic vandalism and their fixes.
- Describe which map feature categories are affected.
- Analyze the spatial extent of cartographic vandalism.
- Identify and characterize users who (1) vandalize OSM or (2) fix vandalized content.

The remainder of the paper is structured as follows. The following sub-sections describe the relationship between PGO and OSM and discuss related work on cartographic vandalism. The OSM data model, the methodology to identify harmful edits related to PGO and the final datasets are described in Section 2. Section 3 presents analysis results, including the evolution of PGO vandalism over time; user groups associated with vandalizing and correcting bad edits, respectively; and the types of features affected by vandalism. Section 4 discusses the findings and the limitations of the study, which are followed by conclusions and directions for future research in Section 5.

*1.2. Pokémon GO and OpenStreetMap*

Released in 2016, PGO is an augmented reality and location-based smartphone game that requires players to navigate to certain locations on a map interface (Figure 1a). Once reaching these points of interest, players can "catch" virtual characters called pokémon with their phone's camera [11]. The goal of each player is to "catch" all available pokémon that appear in the game. PGO was found to increase the physical activity of players [12,13] which can result in measurable health benefits [14]. It was also argued that PGO increases its players' understanding of space and geography [15,16]. These benefits of PGO are closely linked to its location-based nature; that is, it relies on a world map and requires people to visit locations physically. Therefore, the underlying map is of essence for the game. The popularity of location-based games, thus PGO, can be explained by a combination of traditional motivations behind games, such as escaping from a daily routine, and geographic motivators, such as the exploration of new areas [17]. In May 2016, PGO replaced Google Maps with OSM as the background map in the app.

Pokémon do not appear randomly in space but their locations are generated by PGO's proprietary location selection criteria [11]. Since the early days of PGO, anecdotal evidence and community observations suggested that the selection and positioning of PGO features were tied to OSM features. That is, PGO utilizes OSM data (e.g., landuse polygons) to generate pokémon locations in the game [18]. It was not too long until PGO players realized that OSM could be edited, which they turned to their advantage. By editing OSM and planting fictional information, players can gain benefits in the online game. For example, creating a fake lake inside an apartment complex would trigger the creation of "aquatic pokémon". In this example, the player would be able to collect these pokémon without leaving home, which would give them an advantage over other players. Another example is increasing the density of pedestrian features in OSM, e.g., by adding a large number of walking paths to the map (Figure 1b). Apart from gaining benefits in the game, an alternative explanation for PGO vandalism might be that those users are not aware of the purpose of the OSM project, and therefore they do not realize the implications of adding fake data. It is also important to note that only a small percentage of PGO players vandalize OSM, and that some PGO players also actively contribute as mappers within the OSM community.

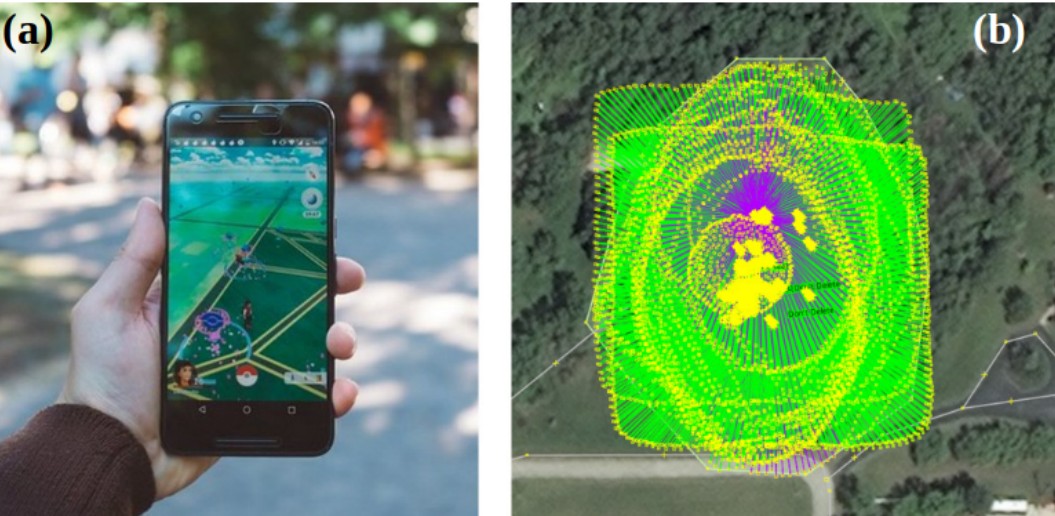

**Figure 1.** (**a**) Smartphone running Pokémon Go (PGO) with its map interface and (**b**) a PGO user adding fake footpaths to OpenStreetMap (OSM) in a park. Image credits: (**a**) https://paintimpact.com/ CC-BY, (**b**) https://github.com/mapbox/mapping/issues/259.

## 1.3. Related Work on Cartographic Vandalism

Digital vandalism is a well recognized problem in online, collaborative settings. In connection with Wikipedia, the world's largest collaborative encyclopedia, vandalism has been studied since the mid 2000s [19,20]. Vandalism and vandal users can be categorized in different ways depending on the research context. However, misinformation and offensive content are often mentioned as their own vandalism category [19,21]. Naturally, this is in-line with carto-vandalism, which is a special case of digital vandalism. Ballatore [5] established the typology of carto-vandalism after reviewing mailing lists and forums related to WikiMapia and OpenStreetMap. The author coined the categories of play, ideological, fantasy, artistic, industrial and spam. Another study used a more quantitative approach and analyzed 51 previously identified cases of vandalism in OSM [22]. It found that 33% of malicious acts added fictional data and that more than 75% of vandalism was committed by new users. Understanding the motivations behind such cases has also been of interest since it can potentially help with detecting them. Coleman et al. [23] identified mischief, agenda (beliefs) and malice and/or criminal intent (personal gain) as main motivations behind carto-vandalism, which matches what has been found for Wikipedia. However, it was argued that this list cannot capture the diverse motivations behind vandalism in a geographic context. When analyzing vandalism in Wikimapia and OpenStreetMap, other motives, such as frustration, boredom, humor or self expression were also identified [5]. Bans in OSM are issued by project administrators for violating community standards. Studying 1218 OSM bans, one study identified 12 common themes for which bans were issued and found that vandalism and politically motivated edits were among the most common types [24].

According to Linus' Law, the collaborative nature of VGI ensures that all vandalism will be discovered and corrected [25]. However, it is unrealistic to expect that all harmful contributions will be found by the mapping community [26]. In reality, the community wastes time and effort finding and fixing harmful edits. As a result, significant effort was put into developing detection systems to identify malicious content. In total, 75% of academic research on Wikipedia was conducted in the computer science field with a focus on the detection of vandalism [27]. Similarly, prototypes of rule- and clustering-based systems were explored for OSM [22,28], and the mapping community is actively using similar systems that aid vandalism detection [29]. Mapbox, one of the most prominent commercial providers that utilizes OSM data uses a hybrid approach that relies on both automatic detection and human review to identify and flag harmful edits, and they also make their detection available for the mapping community [30]. Through this operational detection system, it is estimated that only 0.2% of OSM edits are vandalism [30], which is relatively low compared to the 3–5% that were found for

Wikipedia when analyzing edits to 174 random articles [31]. The direct interaction between PGO and OSM was explored in one study which found that the number of OSM contributors and contributions in South Korea spiked shortly after the introduction of PGO, which implies that PGO players are engaged in editing OSM from the beginning [32]. In the same study, a questionnaire conducted among PGO players who edit OSM revealed that they were primarily motivated to contribute to OSM because of PGO. They expressed the desire to improve the in-game appearance of the map and also to influence the pokémon that appear in mapped locations. The survey also revealed that players who added to the map in OSM because of PGO were more likely to map parks and water bodies when compared to other OSM mappers. Our initial work and the importance of PGO vandalism were also highlighted in a conference presentation [33].

## 2. Methodology and Data Description

### 2.1. OpenStreetMap Changeset and Data Model

The OSM data model consists of three basic map feature types: nodes, ways and relations. Each feature has a unique ID and information about the last user who edited it, the timestamp of the edit and the ID of a changeset the feature was last contained in. Descriptive information about a feature is stored as a set of key-value pairs that are referred to as tags. Geometry is stored for nodes only (as a latitude/longitude coordinate pair) but can be reconstructed into ways, which are ordered lists of nodes. Relations can be constructed from other nodes, ways and relations. Once a feature is edited, its version number is incremented [34]. Changes are not explicitly stored in OSM but can be reconstructed by comparing subsequent versions of features [35]. A changeset is a collection of map edits made in one editing session by a specific user. In addition to newly created features and new versions of edited features, a changeset also contains other mandatory and optional tags with information about the changeset (editor software, mapper, spatial extent, etc.). Although not mandatory, changesets often contain a free text description field (comment tag) to summarize the changes within. As a community standard, editors either make it mandatory to submit this description or strongly encourage users to do so (e.g., with warning messages). As a way to respond to vandalism and mapping mistakes, map edits can be rolled back (reverted) to an older version of data. This is an advanced process that is typically performed by experienced members of the OSM community. Change rollbacks are also map edits that are performed in revert changesets that share the same properties as regular changesets. Revert changesets are usually performed with advanced editor software or a specialized tool (e.g., plugin or script).

### 2.2. Identification of Links between Harmful Edits and Their Fixes

Our approach relies on OSM changesets and changeset descriptions to identify PGO-related carto-vandalism. That is, we first identify changesets that fixed PGO vandalism as a starting point. In a next step the vandalism event is identified by analyzing the content of the fixing changeset. The advantage of this approach is that it uses the OSM community's judgment to find vandalism, as opposed to relying on any heuristic criteria. For this purpose, we consider a changeset only as PGO-related vandalism if it is considered as such by at least one member of the OSM community who makes the effort to fix it.

A typical changeset comment for these fixes is as follows: *"Revert 3243554, Pokémon-Edits"*, which indicates that changeset #3243554 was reverted, and the reason for the revert was that it contained *"Pokémon-Edits"*. Since revert changesets share common characteristics with regular changesets, they contain references to the features that were originally vandalized. Revert changesets are identified based on the changeset's comment tag. More specifically, we construct an ordered list of lexemes from the user-created free text field entry and run a full-text search query for the occurrence of predefined keywords [36]. Lexemes are defined as the minimal units of language [37]. For example, words such as fixed, fixes, fixing, and fix will be represented by the lexeme "fix". Changesets are flagged as fixes

only if the original comment field contains at least one variation of the remove, revert, delete, or fix lexemes and a variation of PGO-related lexemes (e.g., pokemon, pokémon).

Once a revert changeset is identified, we utilize two different approaches to identify the vandalism event. Initial data exploration revealed that the community often indicates the changeset ID of the vandalism it fixed and therefore explicitly creates a link between a vandalism and its fix. We use regular expression (regex) searches for 8-digit numbers that correspond to changeset IDs. The advantage of this approach is that it will establish links between fix and vandalism regardless of how the vandalized changeset was mentioned. For example, comments *"Fixed bad pokemon edits in https://openstreetmap.org/changeset/12345678"* and *"Reverted vandalism in changeset #12345678"* will both establish a link between a fix changeset and vandalism in changeset 12345678. Figure 2a illustrates this process and also highlights that a fix can be linked to more than one changeset. If such a link cannot be determined from the changeset comment, vandalism can be identified through the history of feature edits contained in the fix changeset. In other words, the vandalism changeset is extracted from the previous version of features (Figure 2b). Both methods of identifying PGO-related vandalism result in an underestimation of the phenomenon since (1) not all harmful edits are discovered by the community, and (2) not all fix changesets provide a comment tag that ties the reversion to PGO. However, the method identifies a sufficient number of vandalism changesets that can be used to improve our understanding of carto-vandalism.

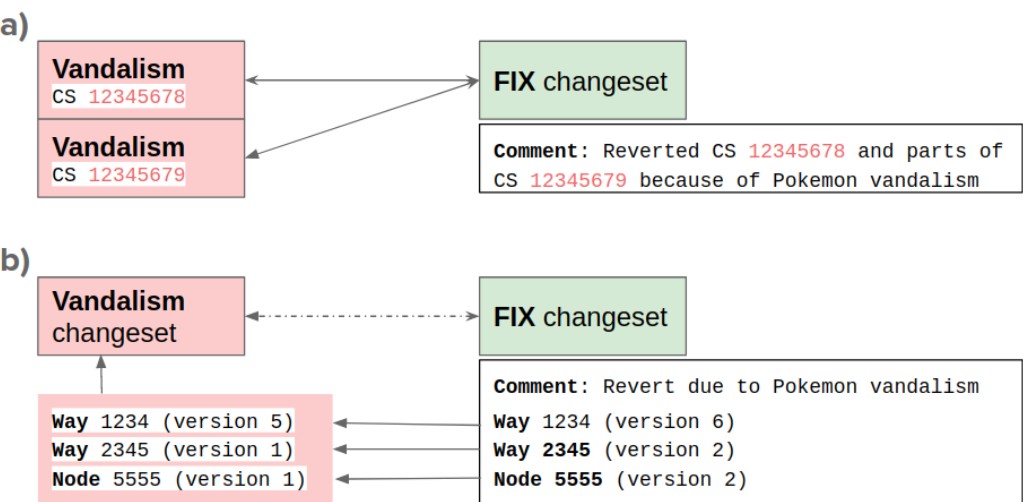

**Figure 2.** Identifying links between vandalism and fix; (**a**) through regex search, and (**b**) through extracting changesets from feature history.

A changeset dump generated on 30 December 2019 from https://planet.openstreetmap.org was used to identify vandalism and fix changesets. This file contains all changesets ever made to OSM. A modified version of the ChangesetMD (https://github.com/jlevente/ChangesetMD) tool was used to insert the changesets in a spatially enabled PostgreSQL database. Processing steps were performed through a combination of UNIX utility tools, SQL queries and self developed R and Python scripts. The input data file, software and a description of processing steps are provided as Supplementary Materials.

### 2.3. Supplementary Data

To assess the experience levels of users engaged in PGO vandalism or their fixes, respectively, we extracted user information through the OSM main API [38] for all users associated with the changesets that were flagged as either fixes or vandalism, such as time of account creation, or number of changesets made. This data was collected on 2 January 2020. The OSM "age" of a user

can be defined as the time elapsed between registering the OSM account and submitting a changeset. Even though account age may overestimate a contributor's experience [39], it is an important proxy for user behavior, which can be an important differentiation criterion for future vandalism detection systems. To better understand vandalism it is important to obtain information about which features PGO players changed in OSM and in which way. For this purpose we extracted Augmented Diffs (adiff) through the Overpass API [40] for each fix and vandalism changeset, which compares the OSM database between two different time instances. For our purpose, we queried the database for changes between the time a changeset was opened and closed, using the timestamps determined by the OSM server, for the areas that matched the changeset. This step facilitates analysis of the content of vandalized changesets. Each feature was assigned a feature category based on the default JOSM preset. This preset establishes a hierarchy of OSM feature categories by automatic comparison of their tags to a predefined list of tag combinations. Similar map features, such as highways, facilities, geography, sport, etc., are grouped into top level categories, with categories further redefined in subsequent levels. Although there are other ways to classify OSM features, we consider the default JOSM preset as a widely accepted categorization. Figure 3 illustrates this preset as it appears in the JOSM editor.

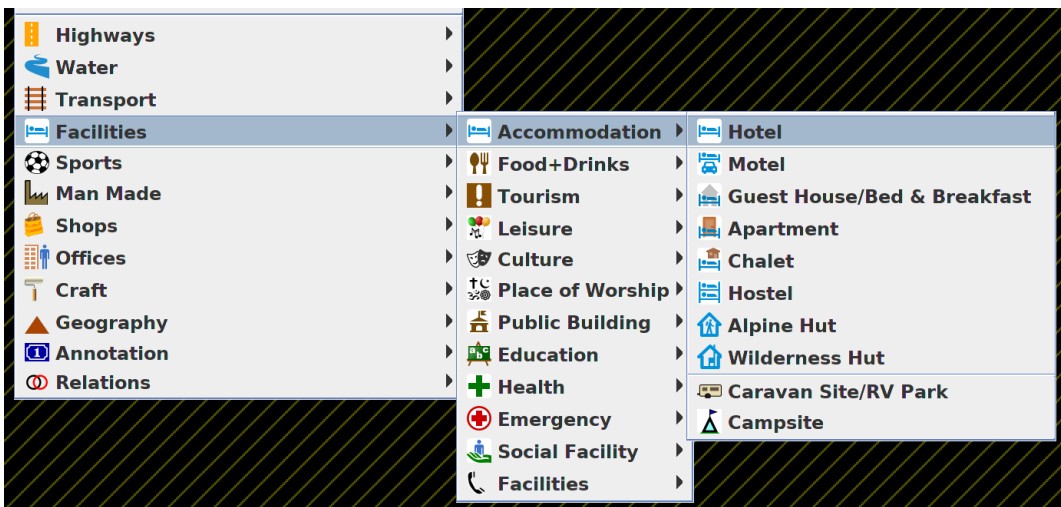

**Figure 3.** An example of the default JOSM preset with different category levels.

### 2.4. Data Description and Analysis Methods

The final dataset contains 2280 links between 2058 PGO vandalism changesets and 1410 changesets that fixed them. Vandalism changesets contained 46,219 map changes affecting 10,543 map features. By map features, we refer to features with attribute information that is indicated within a tag (i.e., simple nodes as parts of ways are not counted). Table 1 summarizes statistics for both types of changesets within the available dataset. A few user accounts were deleted between the vandalism act and our data collection campaign in January 2020. Figure 4 shows the spatial distribution of vandalism changesets by plotting changeset geometries aggregated by their 3-character long geohash representations, which corresponds to a grid resolution of about 0.7 degrees. The countries most affected by PGO vandalism were Germany, the United States and The Netherlands, followed by Taiwan and the United Kingdom.

**Table 1.** Summary statistics of vandalism and fix changesets.

| Type | # of Changesets | # of Users | # of Deleted Users | First Occurrence | Last Occurrence |
|---|---|---|---|---|---|
| Vandalism | 2058 | 815 | 18 | 2016-06-19 | 2019-12-24 |
| Fix | 1410 | 174 | 5 | 2016-07-11 | 2019-12-24 |

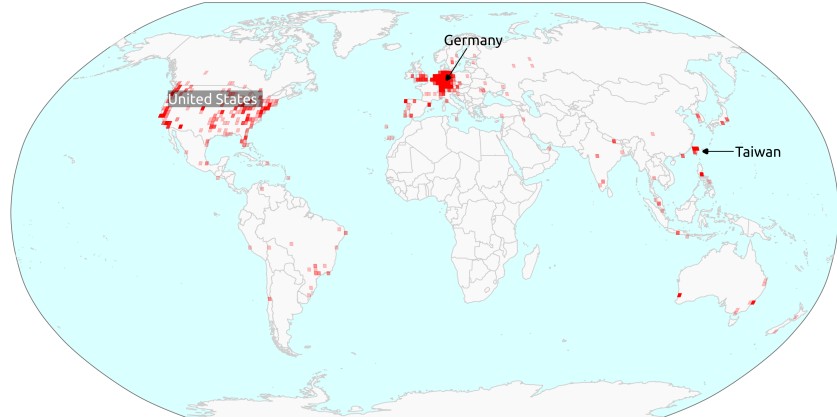

**Figure 4.** Spatial distribution of PGO vandalism.

To assess the temporal dynamics of PGO-related vandalism events, first, the number of changesets and users, separated into fix and vandalism, were aggregated by month. In addition, the time difference between a vandalism changeset and the corresponding fix changeset was calculated. This provides insight into the time it took for the community to discover and fix a malicious edit, and this is described with the complementary cumulative distribution of vandalized changesets that needed a given amount of time to get fixed.

Home areas of fixer users were computed to gain insight into whether they are involved in fixing vandalism locally or over a larger area. We define the home area of a user as the area within which the majority of his or her contributions are found [41] and adapt the methodology described in [42], using changeset centroids to compute home areas for each fixer user. The process is illustrated in Figure 5 for a randomly selected user, where the locations of changeset centroids are shown in Figure 5a. First, Delaunay triangles are reconstructed from changeset centroids (Figure 5b), each of which provides a finer mesh for an area where the user is more active. In the next step, triangles with a perimeter larger than 15 km are removed. Then, remaining triangles are merged to a multipolygon (Figure 5c). In the last step, disjoint areas that were built from less than 20 centroids were removed to only retain areas with high activity. The remaining (multi)polygon is considered to be the home area for that user (Figure 5d). Figure 5e illustrates home areas of users in the eastern part of the United States and Canada.

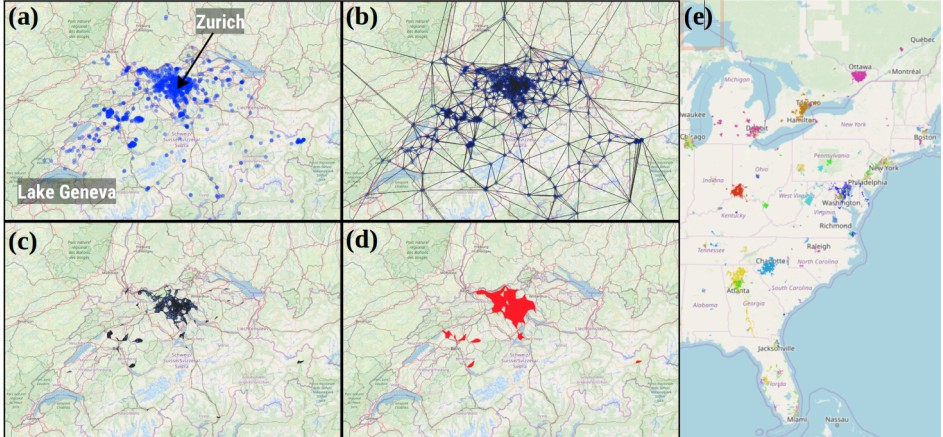

**Figure 5.** Reconstructing home regions. (**a**) Changeset centroids; (**b**) Delaunay triangles constructed from changeset centroids; (**c**) exclusion of triangles with large perimeters, and (**d**) final activity area after merging triangles and excluding continuous areas with only few changesets; (**e**) activity areas of fixer users in the eastern part of the United States and Canada.

Furthermore, the spatial extent within which a user commits PGO-related vandalism around a center location was estimated through the radius of gyration based on each user's list of vandalism changesets [43]. The radius of gyration $R_u$ for a user u was calculated as

$$R_u = \sqrt{\frac{1}{n} \sum_{i=1}^{n} |p_i - c_u|^2}$$

where $n$ is the number of vandalism changesets committed by user $u$, $p_i$ is the centroid of vandalism changeset $i$ and $c_u$ is the center of mass of vandalism changesets for user $u$. Vandalism changesets were found to be small in extent (M = 60.88 km$^2$, MD = 0.02 km$^2$ after excluding the largest 0.5% of vandalism changesets). Therefore, their centroid provides a good estimate of the location of changes included in a vandalism changeset.

## 3. Results

### 3.1. Temporal Characteristics of PGO Vandalism

Figure 6 plots the monthly timeline of PGO-related activity in OSM separated into vandalism and fixes between June 2016 and December 2019. The graph suggests that PGO vandalism did not occur at a constant rate. Instead, clear peaks can be observed in the number of acts of vandalism (Figure 6a) and the number of users engaged in vandalism (Figure 6b) for early 2017 and from late 2017 to mid 2018, which can be attributed to updates of the underlying OSM data and map visuals in PGO. The curves plotting cumulative numbers of users engaged in vandalism and fixes in Figure 6c reveal that the number of fixer users grew at a steady rate and was not as much affected by high activity periods as the number of vandal users was. Figure 6a,b show that there were more vandalism users and vandalized changesets than fixes in almost all months during the entire study's time-frame. The unexpectedly higher number of vandalism changesets in Figure 6a can be explained by the behavior of fixer users who tend to fix more than one vandalism at a time. In fact, on average each fixed changeset fixed 1.5 vandalism changesets and the highest number of PGO vandalism occurrences fixed within one changeset was 24.

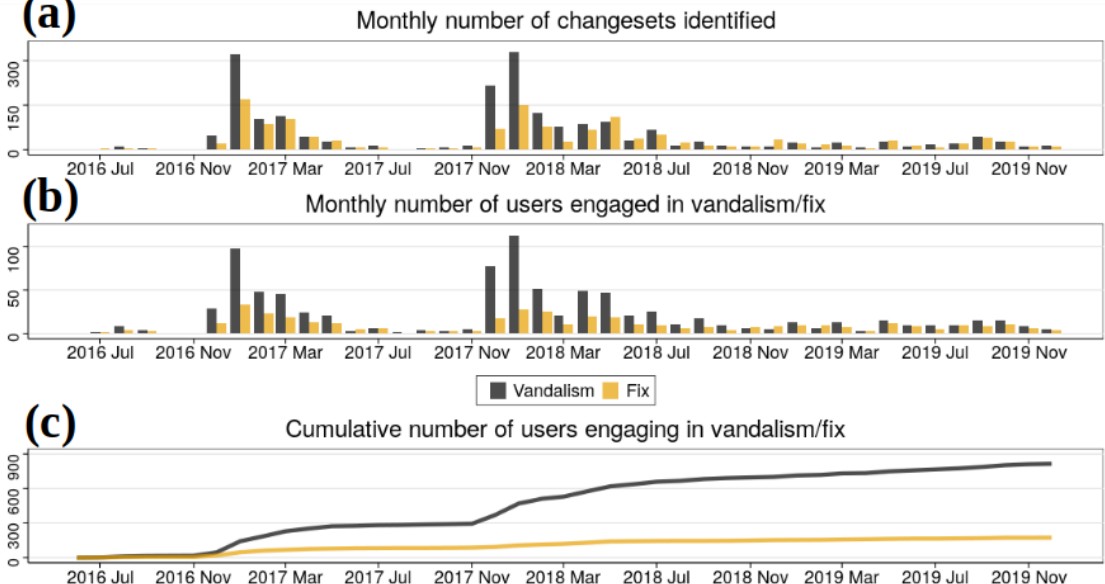

**Figure 6.** Timeline of PGO-related vandalism in OSM; (**a**) monthly number of changesets, (**b**) monthly number of users and (**c**) cumulative number of users.

Ideally, malicious content is instantly detected and fixed by the OSM community. Figure 7a plots the complementary cumulative distribution of time it took to fix vandalism. The figure shows the proportion of vandalized changesets that took more than a given number of hours to get fixed. The long-tailed distribution implies that vandalism changesets are discovered and fixed shortly after they are committed but that there exists a small fraction of long-standing vandalism cases that go unnoticed for a long time. In fact, 11.6% of vandalism was fixed within just one hour, and 65.1% within the first 24 h. Only 16.5% of identified vandalism changesets remained untouched in the system for more than a week. The average time needed to discover and fix vandalism changesets submitted each month decreased over time (Figure 7b). This was confirmed with a Cox–Stuart trend test which revealed a decreasing trend ($p$ = 0.0008). There also appears to be a moderate inverse relationship between the number of monthly vandalized changesets and the time it takes to fix them, as shown in Figure 8. To confirm this, response times during months with a low number of vandalism changesets (less than 50) and during months with a high number of vandalized changesets ($\geq$50) were compared. A Mood's median test revealed that the difference in median values between these two groups is statistically significant ($p$ < 0.0003). This suggests that the OSM community is more alert during time periods of increased vandalism and acts faster than usual.

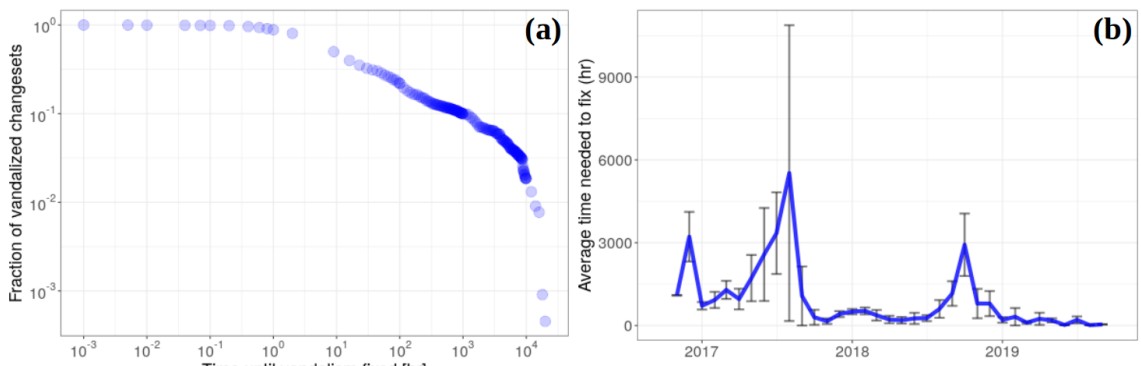

**Figure 7.** Temporal characteristics of vandalism fixes; (**a**) complementary cumulative distribution of time taken by the community to fix each vandalism on a log-log plot, and (**b**) average time required to fix vandalism submitted each month showing a decreasing trend. Standard errors are represented as error bars.

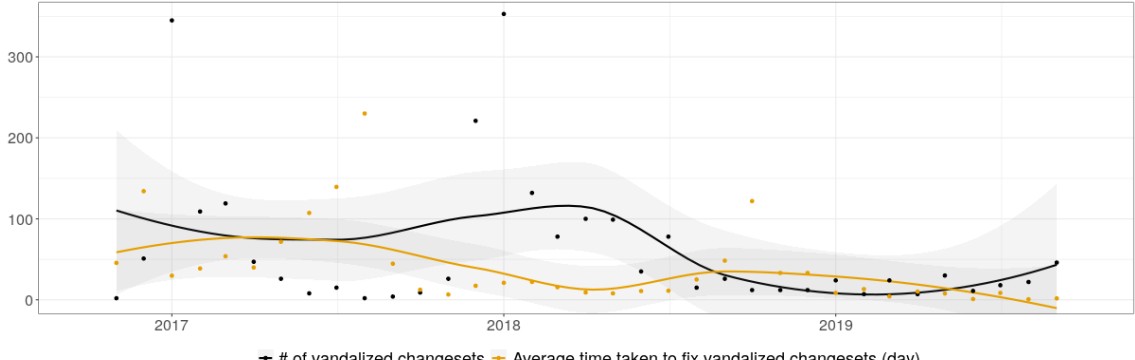

**Figure 8.** Inverse relationship between the number of vandalized changesets and the average time it took to fix them.

### 3.2. Vandalized Content

OSM can be edited with a number of different editors that vary in functionality and the level of expertise required by users. Treemaps in Figure 9 show which editors vandals and fixer users prefer, where the size of each cell is proportional to the share of an editor software. The figure reveals a

clear preference for both vandalism and fixing. Vandalism was mainly committed with the web-based iD editor that accounts for 97% of vandalized changesets (Figure 9a), which is the default editor on the OSM website. Other software used to commit vandalism includes the iOS app called Go Map!! (https://wiki.openstreetmap.org/wiki/Go_Map!!), Potlatch and JOSM. The OSM community preferred JOSM and more specialized tools to fix vandalism, such as the reverter plugin and revert scripts (osmtools: https://github.com/woodpeck/osm-revert-scripts) (Figure 9b). Since the presence of the "revert" lexeme was one part of the set of criteria applied to identify fix changesets, this list may be somewhat biased, which does, however, not affect clear user preferences for iD (vandalism) and JOSM (fix).

PGO vandalism in OSM affects approximately the same number of new and existing features. However, the proportion of created and modified features varies strongly between feature types. That is, many more nodes are created than modified, but more ways or relations are modified than created (Table 2). The table also suggests that PGO vandalism mainly affects OSM ways, whereas relations, which are more advanced data types, are rarely created or modified in PGO vandalism.

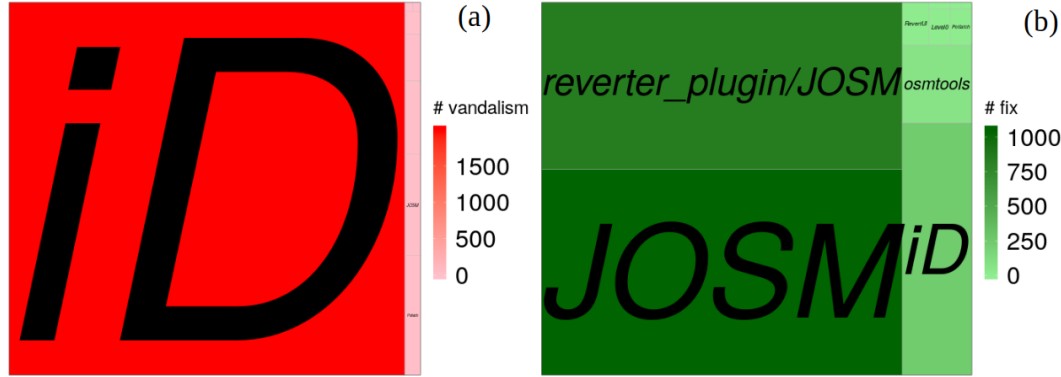

**Figure 9.** Choice of editor software for (**a**) PGO vandalism and (**b**) vandalism fixing.

**Table 2.** Number of map features affected by PGO vandalism.

| Type | Create | Modify |
|---|---|---|
| Node | 1339 | 374 |
| Way | 3910 | 4747 |
| Relation | 28 | 145 |
| **Total** | **5277** | **5266** |

The treemap in Figure 10 shows the classifications of features that were created in connection with PGO vandalism, where feature categories with less than five features were excluded. The size of each cell is proportional to the number of features (nodes, ways and relations) created in that category. Different colors of boxes indicate different main categories, such as facilities or highways. In total, 381 features could not be classified into the default preset, since their tag combination did not match the rules described in the default preset specifications (N/A in Figure 10). These were mainly features with typos in their tags, or with tags not found in the OSM Wiki (see e.g., https://www.openstreetmap.org/way/545937640/history). Half of these features (193) contained only a "name" field but did not indicate the category through a tag. This is a common mistake of new users not familiar with OSM tagging schemes. Geography and highways categories (as per the default JOSM preset described in Section 2.3) accounted for about 60% of all feature creations. Among these, parks (27%) and dedicated footways (15%), in addition to water bodies (5%), dominated new feature creations.

A feature modification can occur as a change in geometry, tag content or membership in relations. For PGO vandalism, tag changes, which include changes in the category of an existing feature,

are the most relevant ones to gain an advantage in the game. An example is the change of a building (building = yes) to a park (landuse = park). Only 6% of all feature modifications resulted in first level category changes in the JOSM preset (e.g., from man-made to geography; see Figure 3), whereas the percentage of modifications that involved a change of the feature category, e.g., grass to park, was much higher. More specifically, 38% (985 features) of feature modifications identified in vandalism changesets resulted in a category change. The Sankey diagram in Figure 11 shows how features were recategorized in vandalism changesets (from left to right), where links with less than five features were excluded for presentation clarity. PGO players tend to recategorize all kinds of outside area features into parks, and various other features into dedicated footways and paths.

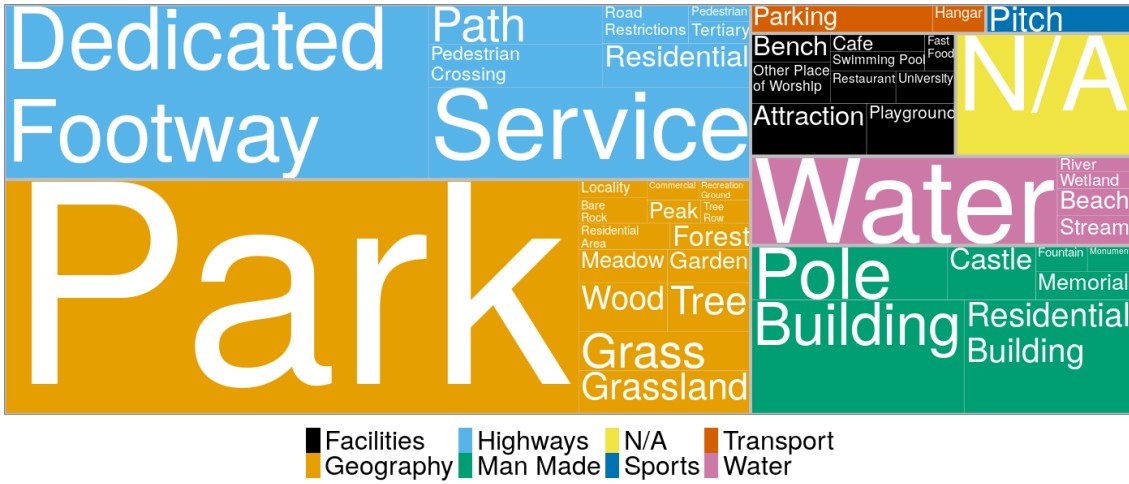

**Figure 10.** Newly created OSM features classified according to the JOSM default preset.

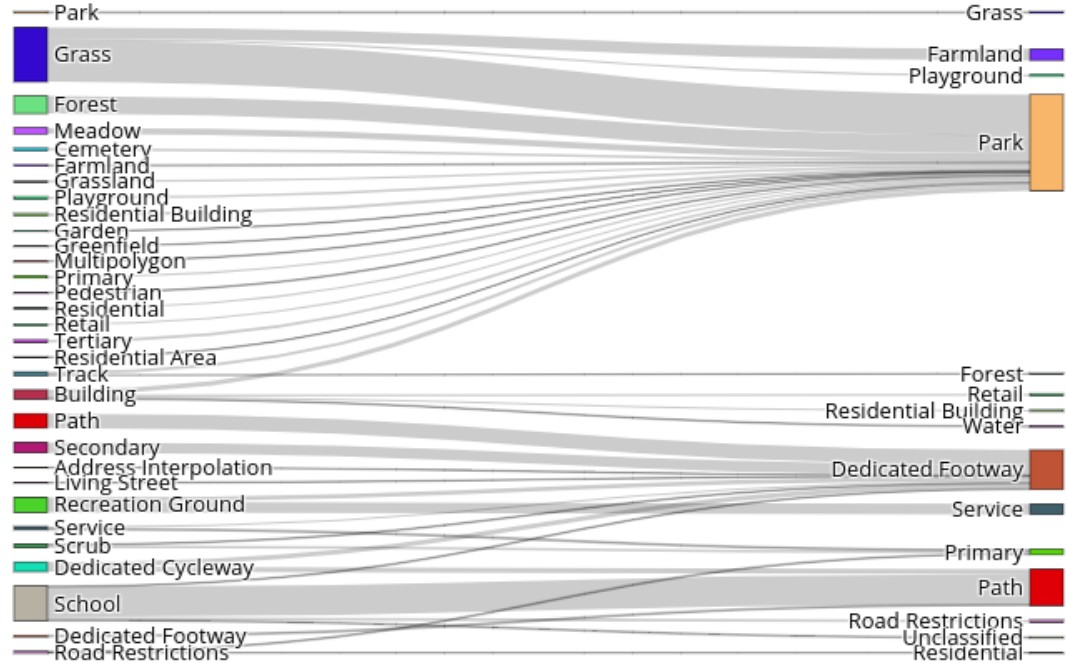

**Figure 11.** Changes in feature categories in PGO vandalism.

### 3.3. User Group Analysis

It is expected that the characteristics of users who engage in vandalism and those who fix these harmful edits are different. The group of 815 unique users who submitted vandalism changesets was more than 4.5 times larger than the user group associated with fixing vandalism (174 unique users). Eighteen vandals and five fixer users who deleted their accounts were excluded from the following analysis. A Mann–Whitney U test revealed that the account age differed significantly between the vandalizing OSM user group (median age = 46 min) and the fixing OSM user group (median age = 4 years and 339 days) ($p < 0.0001$). It was found that 53% of vandal users submitted their first vandalizing changesets within an hour from account creation. At the end of the first month of being a project member, 78% of identified vandals had already made their harmful edits.

To assess the re-occurrence of PGO vandalism activities for a given user, the percentages of first time users and returning users were calculated for both user groups for each month. A user was counted as a returning user only he or she was found to engage in the same type of activity (e.g., vandalizing or fixing) at an earlier time. Returning users were further divided into users who returned only once (i.e., edited in two different months) and users who returned more than once (i.e., edited in more than two different months). Figure 12a,b shows the percentages of user types between July 2016 and December 2019. Figure 12a reveals that the majority of vandals do not engage in vandalism on a regular basis. As opposed to this, fixing vandalism seems to be a sustained activity; the majority of such users engage in fixing harmful edits on a continuous basis on multiple occasions after their first fixes (Figure 12b). Missing bars in Figure 12b are months with no fixing activity.

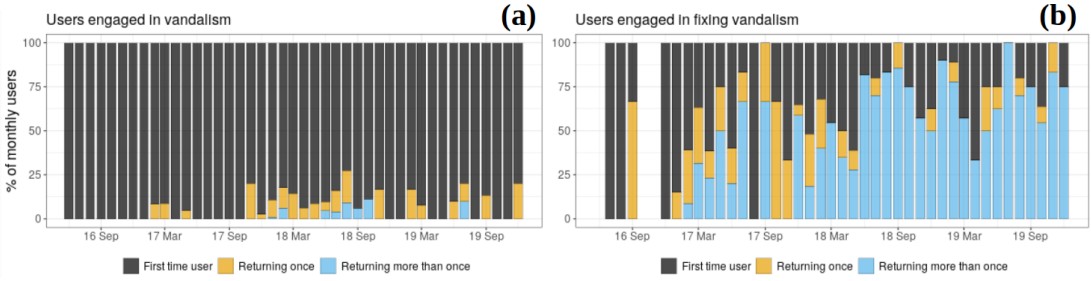

**Figure 12.** Shares of different user types for (**a**) PGO vandalism events and (**b**) for fixes.

Geodetic (shortest path) distances between the home area of a fixer user and each vandalism the user fixed were calculated to reveal whether the community focuses on fixing vandalism only locally or not. Two groups of fixer users can be identified: 46% of all fixer users fixed vandalism only within their home area (i.e., polygons similar to Figure 5d,e). Furthermore, another 10% of fixer users fixed vandalism within less than 100 km from their home area on average. These can be considered local fixers. The remaining 44% of fixer users had a minimum distance of 640 km between their fixes and home regions on average with a standard deviation of 1602 km indicating no clear geographic focus. Users in this group account for fixing almost 75% of all vandalized changesets.

The geographic focus of PGO vandalism was evaluated by calculating the radius of gyration metric for each user found to submit vandalism changesets, which characterizes the spatial extent of vandalized changesets. As expected, the overwhelming majority of users vandalize OSM on a very localized scale, with 94% of vandal users having a radius of gyration of less than 5 km. On the other end of the spectrum, nine users (1.1%) have a radius of gyration value greater than 100 km, implying that their vandalized changesets were spread out over larger areas. Upon examination of these changesets, users in this group still vandalized OSM within one country with one cross-country exception between population centers in the west coast of the USA and Canada, and one cross-continent exception between the USA and Latin-America. The spatial extent of vandalism at either of these locations was small. The distribution of radius of gyration values for vandal users was compared to that of a group of randomly selected OSM users and their changesets, for which the radius of gyration was computed

as well (including all of their changesets). A two-sample t-test was calculated on *log(1 + x)* transformed radius of gyration values (in km), which revealed that the radius of gyration (values given in km) is significantly smaller (M = 28.69, MD = 0, SD = 363.61) for vandalism users than for the general OSM population (M = 452.46, MD = 0.51, SD = 1507.47) (*p* < 0.0001). Figure 13 plots the cumulative percentage of users with radius of gyration values up to a given distance. The plot supports the above finding, as it shows a given percentage of vandal users conducting their edits within a smaller radius when compared to the general mapping population.

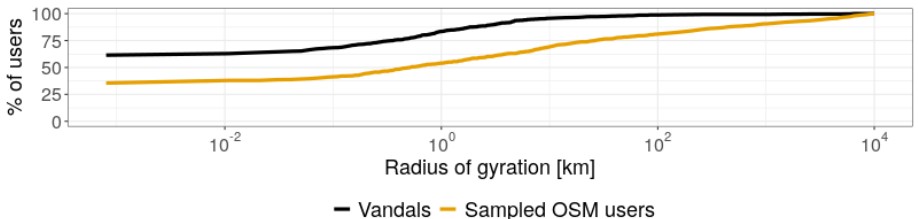

**Figure 13.** Cumulative curves of users with radius of gyration up to a given value.

## 4. Discussion and Conclusions

### 4.1. Discussion of Results

Cartographic vandalism has been in the spotlight in recent years as VGI and other crowdsourced spatial data are increasingly utilized in popular applications. This research described carto-vandalism in a data-driven fashion using Pokémon GO and OpenStreetMap as examples. Our research expands the literature on cartographic vandalism by analyzing its characteristics using a large sample of events. Our analysis results are inline with previous research that analyzed 51 OSM vandalism events [22] and confirms that carto-vandalism affects all world regions. We found that around 65% of vandalism is found and fixed by the community within a day. Similar results for Wikipedia suggest that this pattern might not be specific to carto-vandalism [21,44] but true to digital vandalism in general. Some other patterns also emerged from the analysis, such as that the OSM community seems to get faster at discovering and fixing vandalism events over time. This could potentially be explained by gradually making improvements to existing vandalism detection tools, such as OSMCha and the Find Suspicious OpenStreetMap Changesets website. There also appears to be an inverse relationship between the time it takes to discover and fix vandalism and the number of vandalism events. A possible explanation is that the community pays more attention to this particular type of vandalism when a high number of vandalism cases occurs. Cycles of high PGO vandalism periods (peaks in Figure 6) correspond to periodic map updates of PGO. There have been documented updates between December 2016 and January 2017 and between December 2017 and January 2018 [18]. This implies that, at least for carto-vandalism related to location-based games, creators of games can influence and potentially reduce vandalism from their end; e.g., when pulling dynamic map updates into PGO on a continuing basis. This has also been noticed in a white paper published by PGO community members, who describe their vision of how map data should be incorporated into the game [18]. One of their suggestions is transparency in terms of map updates. Undoubtedly, coordination between the mapping community and data consumers would be useful for battling vandalism. Apart from knowing when there is a planned map update (and therefore an increased number of vandalism events), this potential collaboration could include the disclosure of which map data are being used. Section 3.2 analyzed vandalized map features and showed how certain feature types are more affected by PGO vandalism than others. This is directly related to how PGO uses OSM data, as these features influence the resources available in the game. Currently, the PGO community confirmed a set of OSM tags that increase their game resources in the white paper above. Open communication between game creators, the mapping community and players would therefore (1) eliminate trial-and-error map edits from players when

reverse-engineering the game, and (2) provide the mapping community with tag combinations that could be used in vandalism detection systems.

Unlike other types of online vandals, such as trolls and hackers, whose actions are repetitive [45], PGO players do not sustain their activity over time and typically do not come back to vandalize OSM later, which is due to their different motivations. While trolls and hackers may act out of boredom or based on ideology, it is mischief that drives PGO vandals, whose goal is to gain personal benefits. When this is not achieved, because, for example, their edits are reverted, there is no point in doing it again, which is confirmed in Figure 12a. A portion of PGO vandalism events are also a result of ignorance, since PGO players new to OSM often fail to realize community standards. Our results indicate that PGO vandalism happens on a localized scale, as 94% of edits spread across less than 5 km, which matches the average distance of furthest PGO points a player visits in a day [46]. However, a small portion of vandal users (1.1%) were found to vandalize in multiple distinct locations far from each other. Thus, attention should be paid when describing user behavior with global spatial metrics, such as the radius of gyration. The activity space of PGO vandals is significantly smaller than that of a set of random sample of OSM users. As opposed to this, fixing vandalism shows entirely different spatial and temporal patterns, as it is mainly done by a dedicated group of editors who repeatedly fix vandalism. This group shows similar characteristics to vandal fighters identified in Wikipedia [47]. Furthermore, this group is also more diverse than vandals. Two main strategies were identified based on the spatial patterns of fixes. Nearly half of these users fixed vandalism only within their home area. This is similar to general PPGIS (public participatory geographic information systems)/VGI project participants, who show most effort if their livelihoods are close to the the mapping tasks [48], in this case fixing vandalism. Remaining users fixed vandalism with no clear geographic focus, and often far from their home area. This indicates that their motivation was only to fix vandalism regardless of the geographic area. It is likely that these users heavily utilize existing vandalism detection systems to discover malicious content.

According to a survey among PGO players [32], the ones who constructively edit OSM maps edit exactly the same feature types as those that are affected by PGO vandalism. It is therefore difficult to decide whether an OSM edit is vandalism or not based on content only. This also brings up an important question—whether the popularity of location-based games making VGI visible to massive audiences can be exploited by the mapping community to gain new contributors from this new group of people. This paper did not attempt to answer this question; however, previous research suggests that specific event types (e.g., media articles with massive audiences) can induce a gain in new contributors [39,49]. Specifically linked to PGO, the OSM community already observed a huge gain in Brazil and Portugal after a popular online person stated in a video that players should edit OSM (https://www.openstreetmap.org/user/Jennings%20Anderson/diary/390743#comment45514). It is also important to mention that not all PGO users vandalize OSM and that several members of the OSM community are also PGO players.

The significance of this research can be summarized as follows. It filled an important gap in the literature by describing cartographic vandalism with a data-driven approach and confirmed and extended previous findings on the topic. Previous research either used qualitative approaches or relied on a small sample size. The results of our analysis can be used to improve vandalism detection systems in the future. These results include the characteristics of vandals, the content they add to OSM and also the spatial and temporal distributions of their edits. We expect that the benefits will impact the mapping community in mainly two ways. First, the resources (time and effort) needed to deal with vandalism will be lower as vandalism detection improves. Second, improved vandalism detection will lower the chances of bad data being picked up by third parties and show bad data to a massive audience. While completely avoiding bad data due to vandalism is impossible, the decreased frequency should have a positive effect on the general credibility of VGI.

*4.2. Limitations of the Study*

One limitation of the presented study is that it relies on the OSM community's judgment as to whether a changeset is considered vandalism or not. While it is true that most changesets that were reverted by an experienced member from the community in this context were vandalism, some false positives might have been captured as well; for example, if the reverter did not have local knowledge in an area and misjudged the intent of another mapper. The methodology was also not designed to capture all PGO-related vandalism in OSM. It is expected that (1) PGO-related fixes happened during the study period were not identified by our methodology, and that (2) some PGO vandalism remained undetected and was never reverted by anyone. Despite these obvious limitations, the methodology presented here results in a large sample of vandalism events and their fixes for a detailed quantitative exploration of cartographic vandalism.

Another limitation stems from the fact that this research only analyzes PGO and OSM, which clearly do not represent all location-based games and VGI platforms. As a result, generalized conclusions that apply to all VGI platforms and location-based games might not be appropriate to draw in some cases. For example, results of the content analysis must be handled with care when applied to different applications because they are specific to PGO. Similarly, PGO provides a localized user experience that requires users to interact with their close environment. This is not necessarily a general characteristic of all location-based games. Therefore, the spatial extent of cartographic vandalism is not always limited to local scales. While VGI is similar to other participatory spatial data creation processes, such as PPGIS and citizen science, important differences exist. For example, the practice of PPGIS is by design a slow, careful and reiterative process as opposed to VGI's purpose which is to collect data relatively quickly from a large audience [9]. As a result, PPGIS might be less prone to vandalism than VGI.

## 5. Summary and Future Work

This paper analyzed cartographic vandalism using Pokémon GO and OSM as examples. First, we presented a methodology to identify vandalism events together with their fixes. These links allowed us to study not only vandalism events but the mapping community's response as well. Results are aligned with what has been described for digital vandalism related to Wikipedia, and with the small number of previous studies analyzing cartographic vandalism. These findings can provide ways to improve and fine-tune existing vandalism detection systems, which, in turn, will help maintain the credibility of VGI and save time and resources for the mapping community that battles vandalism. The main findings of this research can be summarized as follows:

- Most carto-vandalism events are discovered and fixed by the community within a day.
- The detection time of PGO-related carto-vandalism gradually decreases over time.
- Individual PGO-related carto-vandalism events are small in extent but affect all world regions.
- The intensity of carto-vandalism is influenced by how VGI data are ingested by location-based games.
- PGO-related carto-vandalism is not repetitive and most users do not vandalize OSM over longer time periods.
- A dedicated portion of the OSM community is engaged in repeatedly fighting vandalism over longer periods.
- Two strategies of fighting vandalism were identified: within one's home area and without a geographic focus.

Although the analysis presented in this paper was specific to PGO and OSM, findings may have implications for the larger realm of VGI and participatory mapping. Arguably, there is a risk of harmful content infiltrating a project whenever a participatory process is involved in the creation of spatial data. However, due to the different nature of participatory processes, the level of threat vandalism poses to different projects has to be further studied. For example, VGI might be more prone to vandalism than

PPGIS or citizen science projects due to its dynamic, more collaborative nature. However, up-to-date data will always be needed as certain events unfold (e.g., disasters) when authoritative sources may not have the resources and procedures to act quickly. Another aspect is the level of detail they may provide. A recent example is the 2019/2020 coronavirus (COVID-19) outbreak, for which the World Health Organization publishes the number of confirmed cases aggregated by country. This aggregation masks the fine spatial variation of spread of the virus. To overcome this, multiple volunteer projects emerged from all over the world that compile detailed maps from various media reports and official press releases. It is not hard to imagine how cartographic vandalism driven by mischief could trigger dangerous mass hysteria or panic if fictional infected cases were reported. For this reason, more research is needed to understand how cartographic vandalism affects participatory spatial projects, and to determine general rules to identify vandalism.

Our future research will be developed in two directions. First, it will explore cartographic vandalism within the larger context of participatory mapping, and in the case of VGI, it will address whether interactions and communication between the mapping community and first time vandals can turn some of these users into valuable members of the OSM mapping community.

**Supplementary Materials:** The following are available online at https://doi.org/10.34703/gzx1-9v95/9MNF1N. Replication Data and Methodology for: Cartographic Vandalism in the Era of Geo-Gaming—the Case of OpenStreetMap and Pokémon GO.

**Author Contributions:** Conceptualization, L.J.; methodology, L.J.; software, L.J. and S.Q.; formal analysis, L.J.; investigation, L.J., T.N. and H.H.H.; writing—original draft preparation, L.J.; writing—review and editing, L.J., T.N., H.H.H. and S.Q.; visualization, L.J. All authors have read and agreed to the published version of the manuscript.

**Funding:** This research received no external funding.

**Acknowledgments:** Authors would like to thank all contributors of the OpenStreetMap project for creating and maintaining an open map database of the world. We are also thankful for the constructive feedback of four anonymous reviewers that improved our work as well as for comments received at the State of the Map 2019 conference.

**Conflicts of Interest:** The authors declare no conflict of interest.

## Abbreviations

The following abbreviations are used in this manuscript:

| | |
|---|---|
| COVID-19 | Coronavirus disease 2019 |
| OSM | OpenStreetMap |
| VGI | Volunteered Geographic Information |
| PGO | Pokémon GO |
| PPGIS | Public Participatory Geographic Information Systems |
| UGGC | User Generated Geographic Content |
| regex | Regular expressions |

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
