# Peer review of "Cartographic Vandalism in the Era of Location-Based Games—The Case of OpenStreetMap and Pokémon GO"

_ijgi, doi:10.3390/ijgi9040197_

Round 1

Reviewer 1 Report

The article "Cartographic Vandalism in the Era of Geo-Gaming -The Case of OpenStreetMap and Pokémon GO" addressed the problem of vandalism in VGI and revealed insightful phenomenon based on thorough analysis. The findings and the methodology should be interesting for VGI researchers.

The article used some innovative ways to address the information insufficiency of VGI research, including the selection of vandalizing & fixing changesets, the determination of user contributing areas, and the analysis schema to reveal what the vandalizing users edited. The statistical methods were generally well-chosen, where multiple robust statistical methods are combined to handle the long-tailed data.

There are several issues which can be improved:

Line 252: Fitting the distribution of fixing time using the power-law function can be inaccurate. Although power-law distributions are reported for VGI in previous literature (Juhász 2016), it may not apply to OpenStreetMap (https://www.mdpi.com/2220-9964/5/1/5) since VGI projects can differ in user contributing patterns (https://www.mdpi.com/2220-9964/9/1/10). Moreover, fitting long-tailed distributions is inherently hard and can be misleading even with high p-values (http://epubs.siam.org/doi/abs/10.1137/070710111). Figure 7a had several problems including 1) using linear regression on log scale, 2) fitting sample distribution instead of cumulative distribution on discrete values, 3) no p-value and confidential intervals 4) the line was not a good fit even visually. Because the fitting brought many problems but didn't add much value to the discussion, it is recommended that the authors simply use boxplots instead to describe the distribution.

Line 257: Linear regression on log-scale can introduce significant bias. Trend test methods specifically designed for non-linear data will be more appropriate.

Line 321: To present the sustaining nature of anti-vandalism actions even better, it is recommended that 1) "returning users" can be further divided into returning-once users and returning-twice-or-more users, and 2) all "fixing" changesets by fixers can be extracted to demonstrate how these contributors work against vandalization in a context wider than the Pokemon Go case.

Author Response

Please find our response in the attached PDF document.

Reviewer 2 Report

Referring to abstract "As a result of OSM’s increased popularity, the worldwide audience ... is directly exposed to malicious edits which represent cartographic vandalism." and after line 35: "That is, VGI projects are vulnerable to cartographic vandalism". This statement hat to generalized since vandalism happenes to other (commercial) mapping projects - specifically to Google Maps - too through their editing and feedback channel.

In line 86 "1.3. Related Work on Cartographic Vandalism" I' dexpect that following talk is also being cited: "Can we validate every change on OSM?", Lukas Martinelli, State of the Map 2018, Milan:
https://2018.stateofthemap.org/2018/T079-Can_we_validate_every_change_on_OSM_/ .

Author Response

(The authors gave the same response as above.)

Reviewer 3 Report

The article offers an analysis of vandalism events linked to the use of VGI (OpenStreetMap) data in a popular geo-game (Pokémon Go). It describes the nature of these vandalism events and presents the response from the VGI community. The authors analyze the changes made to the data on both sides and characterize those who made these changes (vandals and correctors).

Overall Comments

Overall, the document is well constructed. The literature review is relevant. The methods and analysis seem scientifically sound. The text can be further improved to clearly state the objectives of the authors and make the reader aware of the analyzes and results to come.

Throughout a paper, the reader must be able to make the link between the objectives stated in the introduction, the analyzes carried out and the results obtained. These links are not (clearly) made in the text, which makes the reader lose interest in the paper. Make sure that the links between the research objectives, the methods used and the results obtained are clearly stated all over the text. The work you have done worth it.

Major Comments

Lines

Comment

215-216

Rephrase. Research objectives did not mention the interest of studying the temporal dynamics of vandalism. See overall comments

217-221

See overall comments about links between objectives, analysis and results

311-312

How do you know these users deleted their accounts? Although possible, it seems very unusual [1]

313-317

If “age” refers to the editing experience of a contributor, you should rather use the time elapsed between their first and latest contribution. The registration date may overestimate contributors’ experience, especially if they registered before 2010 [3].

329-335

Interesting findings, but you should prepare the reader first. See overall comments about links between objectives, analysis and results.

362-364

Interesting findings. A thought: According to your results and the scientific literature you have read so far, do you think this correction speed ratio (65/35) for vandalism is different from the correction speed ratio for casual editing errors? And therefore, if this correction speed ratio (65/35) is the same, could it be considered specific to the Linus’ law?

408-410

Since you consider that question important, Bégin et al. [3] have shown that many types of events may make VGI visible to massive audiences and induces a gain in new contributors. Specifically regarding OSM and Pokémon GO, look at the first graph and the first comment of the “Contributor Lifespans” section of Jennings Anderson’s OSM diary [4]

Section 5

Integrate with Discussion and Conclusion section, remove duplicated statements

Minor Comments

Lines

Comment

63

Add parenthesis (Figure 1a)

138

Add parentheses (e.g. with warning messages).

146

“we first identify changesets that fixed PGO vandalism as a starting point.” Clever!

177-179

It has already been said.

180

What does the term discussion refer to here? Changeset comments? Clarify.

182

Idem

183

“ChangesetMD tool”: add a reference and briefly explain the objectives of its use

198

“JOSM preset” Wouldn’t “JOSM pre-defined tag values” be any clearer

201-202

Interesting, pragmatic point of view.

206

“Edges” Is there a better word to express that relation?

Table 1

“# of users deleted” Needs an explanation (see major comments). Furthermore, in line 311, you mention that 5 fixer users deleted their account. These users are not shown in Table 1.

Figure 7

Power Law Distribution: it may be good, but it could be tricky [2]

275-277

This was important to state, thank you

278-279

Clarify/rephrase

Figure 10

Nice Figure, very informative

304

“edges,” same comment as before

Figure 11

Nice Figure, very informative

327

Unclear. Maybe having both types of users in the same graph would help? You decide.

341-344

Something expected considering 94% of vandalism is made within a 5Km radius. Not sure that describing the extent of actions from about 1% of vandals is pertinent.

386-387

Interesting

407

“ …PGO vandals identified as identified in our study.” Rephrase

444

“Manuscript” be replaced by paper?

References

[1] https://wiki.openstreetmap.org/wiki/FAQ#How_can_I_close_my_account.3F

[2] Clauset, A., Shalizi, C. R., & Newman, M. E. (2009). Power-law distributions in empirical data. SIAM review, 51(4), 661-703.

[3] Bégin, D., Devillers, R., & Roche, S. (2017). Contributors’ enrollment in collaborative online communities: the case of OpenStreetMap. Geo-spatial information science, 20(3), 282-295

[4] (https://www.openstreetmap.org/user/Jennings%20Anderson/diary/390743

Author Response

(The authors gave the same response as above.)

Reviewer 4 Report

This is a well-developed and clearly written paper about a niche application of VGI that is intriguing in its revelations about human behavior.

Overall, the writing crisp and easy to understand. The paper does make some assumptions about the readers' knowledge of some background, including changesets, Linus' Law, tree maps, and basic concepts about PGO (e.g., how does a player win?). The content is sufficiently interesting to attract readers without background in either PGO or open source development.

A few minor notes about the writing: Is carto-vandalism a word or jargon? In line 54 - reminder should be remainder. In line 59 - is should be are. I do not think that Pokémons is the plural of Pokémon, but please check with a source that is more authoritative than me.

Line 109 writes about wasted time as a bad thing. Is that a judgement call? Or are we sure it is a positive thing?

Starting out, it was unclear to me whether the paper made a sufficient argument that it is addressing a problem sufficiently important to warrant a paper. How big a problem is it? Is bad VGI data really such a bad thing? I do think the paper is warranted, but it should establish its importance better.

The paper provided some background on VGI, but could have used a little more to show this as an area of theory and not just applications. Building on that, is it possible that the participatory nature of PGO is different than simple VGI? I think an extra paragraph or two on PPGIS and participation might help readers positions this (which I think is still quite new for them). 

Overall, the use of graphics is strong. The maps in Figure 5 should identify their locations, otherwise some readers will be looking for Lake Geneva in the Eastern US. Figure 9 should be labelled to explain the type of infographic that is being used (even though it has been used before).

Since this paper is nearly ready for publication, I want to focus on one change that would improve it and broaden its audience. Currently, the final conclusions focus fairly tightly on the implications for PGO, OSM and VGI. While it might be a bit speculative, Section 5 could be broadened to reflect what this research suggests about mischief, motive and cartographic vandalism in the larger realm of VGI. Do popular applications like PGO threaten to bring a non-carto crowd into our previously closed world? Is self-interested mischief a threat to PPGIS applications in land use planning or could cartographic vandalism undermine VGI applications in mapping the spread of COVID-19?

This seemingly little paper is on the verge of some large questions, and I think it would be good for it to start to ask them, even if it doesn't have the answers.

Author Response

(The authors gave the same response as above.)
